# Why Is the Correct Selection of *Trichoderma* Strains Important? The Case of Wheat Endophytic Strains of *T. harzianum* and *T. simmonsii*

**DOI:** 10.3390/jof7121087

**Published:** 2021-12-17

**Authors:** Alberto Pedrero-Méndez, H. Camilo Insuasti, Theodora Neagu, María Illescas, M. Belén Rubio, Enrique Monte, Rosa Hermosa

**Affiliations:** Institute for Agribiotechnology Research (CIALE), Department of Microbiology and Genetics, University of Salamanca, Campus de Villamayor, C/Duero, 12, 37185 Salamanca, Spain; alberto.pedrerom@usal.es (A.P.-M.); hcinsuastia@usal.es (H.C.I.); theodora@usal.es (T.N.); millesmor@usal.es (M.I.); belenru@usal.es (M.B.R.); emv@usal.es (E.M.)

**Keywords:** *T. afroharzianum*, biological control agent (BCA), *Fusarium graminearum*, ACC deaminase, fungal phytohormones, IAA, ABA, ROS, antioxidant activity, drought tolerance

## Abstract

The search for endophytic fungi in the roots of healthy wheat plants from a non-irrigation field trial allowed us to select 4 out of a total of 54 cultivable isolates belonging to the genus *Trichoderma*, identified as *T. harzianum* T136 and T139, *T. simmonsii* T137, and *T. afroharzianum* T138. In vitro assays against the phytopathogenic fungus *Fusarium graminearum* showed that the *T. harzianum* strains had the highest biocontrol potential and that T136 exhibited the highest cellulase and chitinase activities. Production patterns of eight phytohormones varied among the *Trichoderma* strains. All four, when applied alone or in combination, colonized roots of other wheat cultivars and promoted seed germination, tillering, and plant growth under optimal irrigation conditions in the greenhouse. Apart from T136, the endophytic *Trichoderma* strains showed plant protection capacity against drought as they activated the antioxidant enzyme machinery of the wheat plants. However, *T. simmonsii* T137 gave the best plant size and spike weight performance in water-stressed plants at the end of the crop. This trait correlated with significantly increased production of indole acetic acid and abscisic acid and increased 1-aminocyclopropane-1-carboxylic acid deaminase activity by T137. This study shows the potential of *Trichoderma* endophytes and that their success in agricultural systems requires careful selection of suitable strains.

## 1. Introduction

Endophytic fungi are important components of the microbiota living in healthy plant tissues without causing them obvious symptoms. They are increasingly being studied due to their ability to assist the plant in its defense against both biotic and abiotic stresses [1]. *Trichoderma* species are common soil residents and most strains have demonstrated the ability to colonize the rhizosphere and to establish symbiotic relationships with plants [2,3], becoming endophytes [4,5]. Some *Trichoderma* species are even recognized as obligate endophytic biotrophs [6].

*Trichoderma* is the most widely used genus of fungi in biological control. There are almost 400 species of *Trichoderma* [7], and more than a dozen have strains in commercial use as they are, themselves, active matter for biocontrol or are used as a biological solution for plant disease control [8,9,10]. Amongst the most used *Trichoderma*-based biological control agents (BCA) are strains of the species *T. atroviride*, *T. asperellum,* and *T. harzianum* [11,12,13,14]. The taxonomy of *T. harzianum* has been particularly controversial for many years [15,16] and new species with high biocontrol interest have been split off from this species aggregate [17,18]. As a BCA, *Trichoderma* can suppress pathogens through the production of hydrolytic enzymes such as glucanases; chitinases; and proteases (parasitism); secretion of toxic compounds, including volatiles (antibiosis); and competition for a specific niche (nutrients, plant tissues, etc.) [6,13,19,20]. This behavior is much more complex than it seems at first glance, since a given strain often synchronously uses several mechanisms or even displays different mechanisms against different pathogens [21,22]. *Trichoderma* can also exert an indirect biocontrol action through the plant by inducing systemic defenses [10,13,23]. In addition to modulating the plant’s signaling network, mediated by phytohormones [24,25,26], *Trichoderma* strains can produce their own phytohormones, such as auxins, gibberellins (GA), cytokinins (CK), salicylic acid (SA), and abscisic acid (ABA), which increase the complexity of the *Trichoderma*–plant interactions and affect processes such as colonization, host plant growth, and the activation of defenses in stressful situations [21,24,27,28,29,30]. *Trichoderma* can also affect plant phytohormone production and networking by the secretion of enzymes that can modify plant ethylene (ET) levels, such as 1-aminocyclopropane-1-carboxylic acid (ACC) deaminase (ACCD) [31].

Wheat is one of the most important crops to humankind with a global production volume of over 772 million metric tons in 2020/2021 [32]. By 2050, as consequence of increasing population growth, feed grain demand is anticipated to reach 880 million metric tons. These crop yields can be affected by pathogen attacks. *Fusarium graminearum* (*Fg*) is a major causative agent of cereal Fusarium head blight disease [33] since it causes yield losses that are derived from reduction of grain size, weight, and germination rates, because of accumulation of mycotoxins in the grains [34]. *Trichoderma gamsii* T6085 has been shown to be effective in reducing wheat colonization and mycotoxin production by *Fg* [35]. Wheat cultivation is also globally affected by water shortages with drought being recognized as an important constraint for wheat production [36]. In addition, wheat is one of the field crops that is most dependent on the application of chemical inputs of both fungicides and fertilizers [37,38,39]. It is evident that the restrictions of the use of agrochemicals and the social demand for sustainable and less polluting agriculture make *Trichoderma* a viable and acceptable alternative.

Reactive oxygen species (ROS) are constantly generated at basal levels by plants as unwanted byproducts of aerobic metabolism. They do not cause damage because ROS are continuously scavenged by the plant’s antioxidant machinery [40]. The delicate balance between ROS generation and ROS scavenging can be disturbed by stress factors such as salinity, extreme temperatures, and drought. ROS serve as damage markers in plants but they also act as triggers for stress signaling to prevent further damage [10,41]. As an effective rhizosphere colonizer, *Trichoderma* tolerates the ROS in the root environment [42], while at the same time it can activate the plant’s antioxidant machinery to scavenge ROS. This *Trichoderma* trait, together with ACCD production, has proven to be effective in protecting wheat plants against abiotic stresses [30,43,44]. Early studies in cocoa described how *Trichoderma* favors the production of plant metabolites that are associated with increased drought tolerance [4,45].

In this study, we obtained 54 isolates from the endosphere of *Triticum aestivum* of the Berdun variety that were assigned at the genus level. The four that belonged to *Trichoderma* were molecularly identified at the species level. Our aim was to determine whether the phylogenetically very close *Trichoderma* endophytic isolates showed similar biocontrol potential and/or beneficial effects on wheat plants to select them for the most efficient application. To achieve this goal, the four *Trichoderma* strains were included in further assays to analyze their: (i) biocontrol potential against *Fg* through different mechanisms, (ii) phytohormone production profiles, (iii) ACCD activity, (iv) capacity for root colonization of other wheat varieties in single or combined strain applications, (v) effects on wheat emergence and plant growth promotion, and (vi) ability to alleviate drought stress in wheat plants.

## 2. Materials and Methods

### 2.1. Isolation of Fungal Endophytes from Wheat Roots

Fungal strains were isolated from the roots of healthy seven-month-old wheat plants (*Triticum aestivum* Berdun R variety) from a field assay. Once the soil that was attached to the roots was removed, 3.5 g of root system was collected, cut into segments of 1 cm, and washed by shaking at 180 rpm for 20 min, as previously described [39]: twice in 20 mL PBS-S buffer (130 mM NaCl, 7 mM Na_2_HPO_4_, 3 mM NaH_2_PO_4_, pH 7.0, 0.02% Silwet L-77); once in 35 mL of 2% commercial sodium hypochlorite; and three times in 35 mL of PBS buffer. Subsequently, the roots were transferred to a tube with 35 mL PSB buffer, sonicated at 40 kHz for 20 min, and washed in 35 mL PBS buffer. The segments were placed horizontally onto potato dextrose agar (PDA; Difco Laboratories, Detroit, MI, USA) that was supplemented with 300 mg/L chloramphenicol (six pieces per 8.5 cm diameter plate). The Petri plates were incubated at 25 °C in the dark for four days to allow fungal emergence from the edges of the wheat root fragments. A PDA plug was taken from the edge of each emerged fungal colony and separately plated onto PDA to isolate endophytes. The pathogen *Fusarium graminearum* (*Fg*) was provided by the Centro Regional de Diagnóstico de Salamanca (Aldearrubia, Salamanca, Spain). The fungi were routinely grown on PDA medium.

#### Assay of Fungal Drought Tolerance

A total of 54 fungal isolates were tested for resistance to different concentrations of polyethylene glycol 6000 (PEG; Sigma-Aldrich Química S.A., Madrid, Spain). A conidial suspension (200 conidia in 10 μL) of the fungus was used to inoculate each well of sterile 96-well flat-bottomed microtiter plates containing 100 μL of potato dextrose broth (PDB; Difco Laboratories, Detroit, MI, USA), with 0, 10, 20, 30, or 40% (vol/vol) PEG. The microtiter plates were incubated at 28 °C and 20 rpm in the dark for 7 days. For each fungus, an assay was performed using four replicates and the growth results were expressed using the following scale: − (no growth), + to +++ (increasing hyphal growth), and ++++ (completely bushy well).

### 2.2. Fungal Molecular Identification

The total fungal DNA was extracted following the method of Raeder and Broda [46] using mycelium that was collected from a cellophane sheet that was deposited on the surface of a PDA plate, where the fungus was grown at 28 °C for 48 h. The ITS1-ITS4 region of the nuclear rDNA gene cluster, including ITS1 and ITS2 and the 5.8S rDNA gene, was amplified with the primer pair ITS1//ITS4 for the 54 fungal isolates, as previously described [17]. In addition, a fragment of the *tef1**α* gene and one fragment of the *acl1* gene were respectively amplified with the primer pairs EF1-728F (5′-CATCGAGAAGTTCGAGAAGG-3′)//tef1rev (5′-GCCATCCTTGGAGACCAGC-3′) and acl1-230up (5′-AGCCCGATCAGCTCATCAAG-3′)//acl1-1220low (5′-CCTGGCAGCAAGATCVAGGAAGT-3′) for the four *Trichoderma* isolates, as previously described [47,48]. The PCR products were electrophoresed on 1% agarose gels. The amplicons were excised from the gels, purified, and sequenced as previously described [17,30]. The sequences that were obtained were analyzed considering homology in the NCBI database, with ex-type strains and taxonomically established isolates of *Trichoderma* as references.

### 2.3. Antifungal Assays of Trichoderma Strains against Fg

#### 2.3.1. Dual Culture

A total of two sets of dual culture assays were performed. The first was the classical confrontation assay between the *Trichoderma* strains T136, T137, T138, and T139, and the isolate of *Fg*, which were carried out as previously described [49]. The cultures of the pathogen and the *Trichoderma* strains growing alone were used as controls. The colony area of *Fg* was measured after 4 days incubation at 28 °C in the dark, and photographs were taken. Dual cultures were performed in quadruplicate and the results were expressed as the *Fg* colony growth inhibition percentage. A second assay was used to evaluate the antibiotic activity of volatile organic compounds (VOC) that were produced by *Trichoderma* against *Fg*. The effects of VOC that was produced by the strains T136, T137, T138, and T139 were tested towards *Fg* mycelial growth in centrally partitioned (biocompartment) 8.5 cm Petri plates. A 5 mm diameter *Fg* mycelium plug was placed at the center of one side of the partition plates containing PDA, and a 5 mm diameter *Trichoderma* mycelium plug was placed at the center of the other partition containing PDA. *Fg* plates without *Trichoderma* inoculum were used as a control. The inoculated plates were sealed with parafilm and incubated at 28 °C. The experiments were conducted four times and photographs were taken after 72 h.

#### 2.3.2. Growth Membrane Assays

Growth assays on the cellophane sheets and 12.5 kDa cut-off dialysis cellulose membranes were carried out as previously described [49], with some modifications. *Trichoderma* strains were previously grown at 28 °C for 36 h and the diameters of the *Fg* colony were measured after 72 h of incubation at 28 °C in the dark. The results are expressed as the percentage of *Fg* growth inhibition by each *Trichoderma* strain tested, T136, T137, T138, and T139, with respect to the mean colony diameters of *Fg* grown alone. Each strain and condition were tested in quadruplicate.

#### 2.3.3. Extracellular Hydrolytic Activities

A *Trichoderma* conidial suspension (10^5^ conidia/mL) was used to inoculate 250 mL Erlenmeyer flasks containing 120 mL of a synthetic medium (SM) [50] that was supplemented with 2% (wt/vol) glucose (SM + 2% Glu) or 0.5% (wt/vol) cell walls of *Fg* (SM + 0.5% *Fg*-CWs), which were incubated in a rotary shaker at 150 rpm and 28 °C for 5 days. *Fg*-CWs were obtained as previously described [51]. Each strain and culture medium were tested in triplicate. The resulting supernatants were collected in a Buchner funnel and then were 100-fold concentrated by ammonium sulfate precipitation and dialyzed against distillate water for 72 h. Quantitative protein determination was performed by a Bradford assay [52] with bovine serum albumin as a protein standard. Protease, cellulase, and chitinase activities were determined in colorimetric assays, as previously described [53], by measuring the release of azo dye during hydrolysis of azocasein at 366 nm, the release of reducing groups during the hydrolysis of carboxymethylcellulose at 520 nm, and the release of N-acetylglucosamine during hydrolysis of chitin at 585 nm, respectively. Measurements were also performed in triplicate. The total activity corresponded to micromoles that were formed in one minute and specific activities corresponded to micromoles in one minute per milligram of protein.

### 2.4. ACCD Activity

The ACCD activity of the T136, T137, T138, and T139 strains was tested as previously described [30]. Briefly, 100 μL of a conidial suspension (1 × 10^6^ conidia/mL) was inoculated in 10 mL of SM [50], and the cultures were grown at 180 rpm and 28 °C for 4 days. The mycelia were collected and homogenized in 2.5 mL Tris buffer 0.1 M (pH 8.5). The protein concentration in the samples was calculated as described above. The reaction mixture was prepared by adding 25 μL of toluene and 20 μL of 0.5 M ACC to a protein aliquot of 200 μL. ACCD activity was determined in a colorimetric assay by measuring the amount of α-ketobutyrate that was produced by the deamination of ACC at 540 nm. A standard curve that was prepared with α-ketobutyrate (10–200 μmol) was used as reference. ACCD activity corresponded to millimoles of α-ketobutyrate that were formed in 1 h, and specific ACCD activity corresponded to millimoles of α-ketobutyrate in 1 h per milligram of protein. A total of three independent cultures were analyzed for each strain, and activity measurements were also performed in triplicate.

### 2.5. Phytohormone Production by Trichoderma Strains

The strains T136, T137, T138, and T139 were grown in 120 mL of PDB and PDB with 200 mg/L of tryptophan (PDB-Trp) media at 180 rpm and 28 °C for 48 h, and the culture supernatants were collected by filtration in a Buchner funnel. Uninoculated PDB and PDB-Trp media were used as their respective controls. The supernatants were lyophilized and the dry weight was measured. A quantity of 50 mg of lyophilized supernatant was used for hormone extraction, as previously described [54]. The production of eight phytohormones [IAA, ABA, cytokinin dihydrozeatin (DHZ), cytokinin isopentenyladenine (iP), cytokinin trans-zeatin (tZ), SA, gibberellin 1 (GA_1_), and gibberellin 4 (GA_4_)] by the four *Trichoderma* strains was determined as previously described [30]. A total of three independent replicate flasks were analyzed for each strain and culture medium.

### 2.6. Colonization of Wheat Roots by Trichoderma Strains

The quantification of *Trichoderma* DNA in wheat roots was performed by quantitative PCR (qPCR), as previously described [30]. The following six conditions were tested: control (uninoculated), T136, T137, T138, T139, and a mixture of these four *Trichoderma* strains. Briefly, three 10-day-old wheat seedlings that were grown in 8 mL of Murashige and Skoog (MS; Duchefa Biochemie BV, Haarlem, The Netherlands) medium plus 1% sucrose were inoculated with 8 × 10^5^ conidial germlings of *Trichoderma,* that was obtained and counted as previously described [55]. The roots were collected 48 h after inoculation of *Trichoderma* germlings or not (control), washed with sterile water, homogenized under liquid nitrogen, and used for DNA extraction. qPCR was performed in a Step One Plus thermocycler (Applied Biosystems, Foster City, CA, USA), using KAPA SYBR FAST (Biosystems, Buenos Aires, Argentine) and the previously described primer couples Act-F (5′-ATCGGTATGGGTCAGAAGGA-3′)//Act-R (5′-ATGTCAACACGAGCAATGG-3′) [56] and Act-Fw (5′-TGACCGTATGAGCAAGGAG-3′)//Act-Rv (5′-CCAGACACTGTACTTCCTC-3′) [26] to amplify a fragment of the *actin* gene from *Trichoderma* and wheat. The reaction mixtures were made in triplicate, and the PCR conditions and DNA estimation using Ct values and standard curves were as previously described [21]. DNA that was extracted from four independent fungus-plant cultures per condition was tested.

The quantification of *Trichoderma* strains was also carried out by counting the number of colony-forming units (cfu) from one root (ca. 0.25 g) of each wheat plant. For this, the roots were sampled on the wheat plants of the greenhouse assay described below at 21 days after sowing. Each sampled root was washed twice in 20 mL PBS-S buffer by shaking at 180 rpm in a 50 mL tube for 20 min. Serial dilutions of the rhizosphere washing liquid were plated on *Trichoderma* selective medium (TSM) [57]. The plates were incubated at 28 °C and the cfu were counted after six days. In addition, the colonies of the Mix treatment that grew on TSM were then separately plated on PDA for subsequent morphological distinction and *Trichoderma* species assignment. One root of the four wheat plants per treatment was tested.

### 2.7. Trichoderma-Wheat Assay under Greenhouse Conditions

The effect of *Trichoderma* strains T136, T137, T138, and T139, that was applied alone or combined in a mixture on wheat growth promotion or drought stress alleviation was tested in a greenhouse assay. Wheat (Basilio variety) seeds were surface disinfected and seed stratification was performed as previously described [30]. The plant growth substrate consisted of an autoclaved mixture of commercial peat and vermiculite in a 3:1 proportion (Projar Professional, Comercial Projar SA, Fuente el Saz de Jarama, Spain). *Trichoderma* mycelium was applied to the plant growth substrate that was contained in conical pots of 250 mL capacity. Mycelium was obtained from a PDB culture, using 0.5 L flasks containing 250 mL of PDB inoculated with 1 × 10^6^ conidia/mL, and grown at 180 rpm and 28 °C for 6 days. The mycelium was collected by filtration, washed with sterile water, and used to inoculate the plant growth substrate. This assay included six initial treatments and a total of 192 plants (2 plants per pot and 32 plants per treatment), named as follows: uninoculated (control), T136, T137, T138, T139, and Mix. Mycelium that was obtained from 250 mL of PDB was used for inoculation of the 16 pots of each treatment, except for treatment Mix, in which one quarter of each mycelium from each *Trichoderma* strain was mixed and the mixture served to inoculate the substrate that was contained in 16 pots. The plants were maintained in a greenhouse at 22 ± 4 °C, as previously described [25], and watered as needed for 3 weeks. The plants of each treatment that are described above were separated into two blocks: (i) 16 plants per treatment were maintained with optimal irrigation (control irrigation condition), and (ii) 16 plants per treatment were maintained without watering for 2.5 weeks (water stress condition). Then, all plants were watered as needed for 15.5 more weeks. A scheme with the running and sampling time points of this experiment is shown in Figure 1. The experiment lasted 21 weeks.

### 2.8. Physiological Measurements in Wheat Plants

The plant emergence rate was determined at 6 and 9 days, and the results were expressed in percentage. Tillering data were taken on 35-day-old wheat plants (eight plants per treatment and condition). The total shoots of wheat plants were collected at 40 days to determine the fresh weight (four plants per treatment and condition). The plant height and spike weight were recorded on 21-week-old plants (four plants per treatment and condition).

### 2.9. Determination of H_2_O_2_ Content and Antioxidant Activities in Wheat Plants

H_2_O_2_ content was determined as previously described [30,58], using 50 mg of fresh plant material that was taken from 40-day-old wheat plants. The results were expressed as micromoles per gram of fresh weight. Four biological replicates per treatment and condition were analyzed.

Three antioxidant enzymatic activities, superoxide dismutase (SOD), peroxidase (POD), and catalase (CAT), that were related to ROS scavenging were analyzed as previously described [30,59], using wheat leaves that were detached from 40-day-old plants and immediately frozen in liquid nitrogen. Proteins were extracted from 50 mg of plant material homogenized in 50 mM potassium phosphate buffer (pH 7.8), and their concentration was determined as previously described [52]. SOD, POD, and CAT activities were expressed as units per minute per milligram of protein. A total of four biological replicates per treatment and condition were analyzed.

### 2.10. Statistical Analyses

IBM SPSS^®^ Statistics 27 (IBM Corp.) was used for the statistical analyses through an analysis of variance (ANOVA) using Tukey’s and Duncan’s test (the latter for data of antioxidant activities and H_2_O_2_ content) to identify the significant differences among the samples (*p* < 0.05). A two-way ANOVA was used to test for possible interactions between the main effects within each set of data, followed by a mean separation using Tukey’s test (*p* < 0.05).

## 3. Results

### 3.1. Isolation of Endophytic Fungi from Wheat Roots, Selection and Genus Assignment 

Using plant material that was collected from a field assay [39], 768 wheat root pieces of 1 cm length, that were cut from previously washed and sonicated roots, were plated on PDA chloramphenicol and, as a result, 54 isolates showing different phenotypes were selected (Appendix A). The sequence analysis of the ITS1-ITS4 region distributed them into 16 fungal genera (Appendix A). Members of the *Fusarium* genus were the most prevalent among the 54 isolates (37.04%), followed by the genera *Alternaria* (13%), *Monosporascus* (13%), and *Trichoderma* (7.5%); 21 out of 54 were able to grow in PDB with 40% PEG. This percentage of PEG is equivalent to a water potential of −1.76 MPa, and although some of the isolates were affected by this simulated drought stress, it was not a constant for all fungi that were tested. Moreover, no correlation was found between belonging to a fungal genus and growing at a given PEG concentration, suggesting an isolate-dependent property. The four isolates of the genus *Trichoderma* that grew to different extents in PDB plus 40% PEG were chosen for further testing.

### 3.2. Characterization of Trichoderma Strains Endophytes of Wheat 

The *Trichoderma* strains T136, T137, T138, and T139, that were isolated from the wheat root endosphere were molecularly identified at the species level by sequence analysis of the ITS1-ITS4 region, a fragment of ca. 600 bp in length of the *tef1α* gene and a fragment of ca. 850 bp in length of the *acl1* gene. The *acl1* sequence that was obtained for T139 was not clean enough for sequencing, although this did not prevent its identification. The accession numbers of their sequences were deposited in the GenBank and are shown in Table 1. As they had sequences that were identical to those of ex-type strains or representative species that were available in databases, they were identified as: *T. harzianum* T136, *T. simmonsii* T137, *T. afroharzianum* T138, and *T. harzianum* T139.

### 3.3. Trichoderma Strains Show Different Biocontrol Potential against Fg

The biocontrol potential of strains T136, T137, T138, and T139 against *Fg* were tested using different in vitro assays. All *Trichoderma* strains were able to inhibit the *Fg* colony growth in dual cultures (Table 2), although two different antagonistic behaviors were observed. Strains T136 and T139 overgrew the *Fg* colony in four days in co-cultures, while strains T137 and T138 did not (Figure 2A). In addition, strains T136 and T139 differed significantly from the other two strains in their ability to inhibit pathogen colony growth (Table 2). To determine whether VOCs that were produced by these *Trichoderma* strains may reduce *Fg* growth, we confronted both fungi in partitioned Petri dishes (Figure 2B). VOCs that were produced by strain T138 appeared to have antifungal activity against *Fg*, while inhibition was not observed for the other three strains.

To determine the capability of *Trichoderma* extracellular compounds to inhibit *Fg* colony growth, the four *Trichoderma* strains were grown on PDA with cellulose or cellophane membranes on the surface of the culture medium to allow the introduction of *Trichoderma* extracellular compounds into the agar medium. After removing the membranes, the effect of different *Trichoderma* extracellular compounds on *Fg* growth was determined. The results are shown in Table 2. As was the case with the dual culture assay, all strains inhibited the growth of *Fg*, and strains T136 and T139 showed the best results in the membrane tests.

The biocontrol potential of the four *Trichoderma* strains was also evaluated by measuring the extracellular protease, cellulase, and chitinase enzymatic activities in SM + 2% Glu and SM + 0.5% *Fg*-CWs cultures (Table 3). Protease and chitinase activities were detected for the four strains in both culture media, apart from protease activity for T138 in SM + 2% Glu. No cellulase activity was detected in SM + 2% Glu medium. The highest levels of protease activity were detected for strain T136 in SM + 2% Glu. However, the highest levels of cellulase and chitinase activities were detected for strain T136 in SM + 0.5% *Fg*-CWs (*p* < 0.01). A two-way ANOVA showed the effect of the variable “strain” (*p <* 0.001, for both protease and cellulase activities; *p* < 0.01, for chitinase activity) and the variable “medium” (*p <* 0.001, for both protease and cellulase activities; *p* < 0.01, for chitinase activity), and the combination “strain x medium” (*p <* 0.001) for all three activities.

### 3.4. Trichoderma Strains Show Differences in ACCD Activity and Phytohormone Production

To obtain data that were related to plant beneficial effects of the four *Trichoderma* strains, ACCD activity and the production of eight phytohormones were investigated. The specific ACCD activity detected after growing *Trichoderma* in a SM medium for four days is shown in Figure 3. All of the strains displayed ACCD activity, the highest being *T. simmonsii* T137 (*p* < 0.05).

The levels of IAA, ABA, DHZ, iP, tZ, SA, GA_4_, and GA_1_ that were detected in lyophilized *Trichoderma* supernatants from two day PDB and PDB-Trp cultures, compared with those that were detected in the corresponding uninoculated controls, are shown in Figure 4. The production of the eight phytohormones was detected in *Trichoderma*. In comparison with the controls, the production of SA by T138, of ABA by T136 and T139, and of GA_4_, GA_1_ and tZ by T137 and T138, was not detected. IAA production in the four strains positively responded to the addition of Trp to the fungal culture medium. However, a negative relationship was observed in all four strains between the production of tZ or GA_1_ and the addition of Trp to the medium. According to a two-way ANOVA, there was an effect of the variable “strain” and variable “medium” and their combination “strain × medium”, on the production of five out of the eight phytohormones that were investigated (Figure 4). Strain T137 was notable in the production of ABA and iP in both media (*p* < 0.05) and of DHZ in PDB-Trp (*p* < 0.05). Moreover, production of IAA was significantly higher in strains T137 and T139 compared with those that were detected for T136 and T138 in PDB-Trp (*p* < 0.05), while SA production was only detected in T136 and T139 supernatants from this last medium (*p* < 0.05).

### 3.5. Efficient Colonizers of Wheat Roots Even if Applied Together

To evaluate the ability of T136, T137, T138, and T139 strains that were isolated from the endosphere of Berdun R wheat to colonize Basilio wheat roots when applied alone or with all four mixed together, we estimated the proportion of fungal DNA vs. plant DNA from qPCR data that were obtained from 10-day-old seedling roots at 48 h after inoculation of *Trichoderma*. As shown in Table 4, *Trichoderma* was not detected in the root samples from uninoculated plants, while all strains were able to colonize the roots of Basilio seedlings. No differences in colonization were detected among the four strains and the Mix.

The abundance of *Trichoderma* strains in 21 day samples was 2.4 × 10^5^, 0.7 × 10^5^, 4.3 × 10^5^ and 4.9 × 10^5^ cfu per wheat plant root for strains T136, T137, T138, and T139, respectively. No *Trichoderma* colonies were observed in the roots of the control treatment. Regarding the Mix treatment, *Trichoderma* abundance was found to be in a similar range, 1 × 10^5^, to individual inoculations. An analysis of the count data showed no differences among the *Trichoderma* treatments. In addition, considering the colony phenotypes on PDA, the four strains had similar percentage counts in the Mix treatment.

### 3.6. Trichoderma Strains had Different Effects on Wheat Plants

*Trichoderma* strains T136, T137, T138, and T139 were tested in an in vivo assay to study the effect on wheat plant growth promotion and the drought stress alleviation of plants when fungi were applied separately or in combination. The effect of these strains on plant emergence, tillering, and plant growth are shown in Table 5. At 6 days, the lowest emergence rate was recorded for the control treatment, at 46.7% (*p* < 0.05), and the highest rates were 83.3 and 80%, observed, respectively, in the T137 and T136 treatments (*p* < 0.05). However, no significant differences were detected in the emergence rates among treatments at 9 days. This result showed that the application of any of the four *Trichoderma* strains or the Mix to the plant growth substrate advanced the germination of Basilio wheat seeds. As shown in Table 5, data of the tillering rate that were recorded at 35 days also showed that this parameter was significantly increased by the five *Trichoderma* treatments, at least under optimal irrigation conditions. The highest tillering rate value was recorded for the Mix treatment in the absence of water stress. Interestingly, under water stress conditions, tillering was observed in all plants of the T138 treatment, and in none of the T136 treatment. Analysis of the fresh weight data that were recorded in 40-day-old plants showed differences among treatments. Under optimal irrigation, the control plants showed the lowest values compared to those from plants of the other treatments. However, under water stress conditions, no differences in fresh weight were observed between the control and T136-treated plants and they showed the lowest values compared to those of the other four *Trichoderma* treatments. A two-way ANOVA for the fresh weight results showed the effect of the variable “treatment” (*p* < 0.001) and of the variable “irrigation condition” (*p <* 0.001), but no showed significance for the combination of both factors.

The analysis of endogenous H_2_O_2_ content that was recorded from 40-day-old plants did not show differences between any of the *Trichoderma* treatments and the control in plants under the optimal irrigation condition (Figure 5A). Differences were seen between plants that were treated with strain T138, which achieved higher H_2_O_2_ values, and those from the T137 and Mix treatments (*p* < 0.05). Water-stressed control plants showed significantly higher levels of H_2_O_2_ compared to those that were treated with *Trichoderma*, which, in turn, presented significant differences, with the lowest value obtained in plants of the T137 treatment. In addition, the effect of the variable “treatment” (*p* < 0.01) and the variable “irrigation condition” (*p* < 0.01) was detected by a two-way ANOVA. In parallel, no difference was detected for SOD and POD activities between plants of any of the five *Trichoderma* treatments and the control plants under optimal irrigation conditions (Figure 5B,C). The highest levels of CAT activity in unstressed plants were detected for the T136 treatment (Figure 5D). In contrast, differences among treatments were observed for the three enzymatic activities in water-stressed plants. In this sense (Figure 5B–D), plants of the T137 and T138 treatments presented the highest levels of SOD and those of the T137 and Mix treatments had the highest levels of POD (*p* < 0.05), while the lowest values of CAT occurred in plants of the Mix treatment. Differences in the effect of the two variables, “treatment” and “irrigation”, and of their combination, were observed for these three activities by a two-way ANOVA. For SOD, the effect of the variable “irrigation” (*p* < 0.001) and the combination of variables “treatment x irrigation” (*p* < 0.05); for POD, the effect of the combination of both variables (*p* < 0.05); and for CAT, the effect of the variable “treatment” (*p* < 0.001) and the combination “treatment x irrigation” (*p* < 0.05) were detected.

To study the effect of the five *Trichoderma* treatments long-term, the greenhouse assay was maintained for five months and all plants were watered as needed when they reached 40 days. The representative plants and spikes from the different treatments and the two irrigation conditions are shown in Figure 6A,B. No differences in plant height and spike weight values were detected among treatments under the optimal irrigation condition, since *Trichoderma*-treated plants gave values similar to those of the control (Figure 6B,C). However, significant differences in these two parameters were detected among treatments under the water stress condition; this started at 21 days and was maintained for up to 2.5 weeks. Plants of the control and T136 treatments gave the lowest values of plant height, and together with those from the T139 treatment, also presented the lowest values of spike weight. Little variability among replicates was observed for these parameters in plants that were treated with strain T137. The two-way ANOVA for height results showed an effect for the variable “treatment” (*p <* 0.001) and the variable “irrigation” (*p <* 0.001), and for the combination of both factors (*p <* 0.01). For spike weight, the results showed an effect for “treatment” and effect for “irrigation” (*p <* 0.01), but no effect for their combination (*p* > 0.05).

## 4. Discussion

Wheat is one of the world’s most important food crops, but climate change, which is causing rising temperatures and an increase of arid and semi-arid arable lands, is leading losses that are often unaffordable. To cope with problems such as attacks by pathogens such as *Fg*, which takes advantage of plant weakness and drought, we have adopted a strategy that was based on the isolation and selection of *Trichoderma* endophytic strains from wheat. Initially, we isolated 54 culturable endophytic fungi from the roots of healthy plants that were grown in the field under non-irrigated conditions. Interestingly, *Fusarium* spp. isolates were prevalent, making up more than one third of the wheat root endosphere in our trial. Although several species of *Fusarium* are recognized as wheat pathogens, the *Fusarium* isolates that were collected in our study were obtained from the root endosphere of heathy plants. Endophytic isolates of *Fusarium* spp. have been described as useful for controlling root and vascular diseases that are caused by oomycetes and fungi [60]. Thus, these wheat endophytic isolates of *Fusarium* should be further investigated since they may not be detrimental and may even be beneficial for the crop. A total of four endophytic strains of *Trichoderma* were also isolated and assigned to three species [7]: *T. harzianum*, *T. simmonsii,* and *T. afroharzianum*. All belong to the former species complex “*T. harzianum*” [18], a species that has many strains that are used as BCAs [11,12]. Although *T. simmonsii* is a common soil inhabitant [18], a recent report also described it as an endophyte in bell pepper plants [61]. Root colonization ability is often a criterion for selecting *Trichoderma* strains that are beneficial to plants [23], and we observed by qPCR analysis that wheat plants of the Basilio variety are suitable hosts for the four *Trichoderma* isolates that were obtained from the wheat root endosphere of the Berdun R variety when applied alone or in combination to seedlings. qPCR colonization data that were obtained 48 h after 10-day-old seedling inoculation with *Trichoderma* in liquid medium were in line with those obtained by assessing *Trichoderma* survival in three-week-old plants that were grown under greenhouse conditions. Data that were obtained in both experiments for individual inoculations and the Mix treatment showed an absence of negative interactions among the four *Trichoderma* strains, indicating compatibility when mixed.

To compare the antagonistic properties of the four *Trichoderma* strains against *Fg*, different in vitro inhibition tests were carried out. *T. harzianum* T136 and T139 showed the best performances in dual cultures and membrane assays, indicating that mycoparasitism, competition for space and nutrients, and antibiosis may be involved in their biocontrol activity against this pathogen. The capability of *T. harzianum* to antagonize different phytopathogenic *Fusarium* spp. has been previously reported [62,63]. Greater inhibition of *Fg* colony growth on cellophane membranes in comparison to that which was observed on dialysis membranes indicates that the antifungal activity of both strains of *T. harzianum* is due to the combined effects of secreted low and high molecular weight compounds. This result contrasts with previous studies showing that metabolites that are produced by *Trichoderma* spp. are the main contributors to the antifungal activity of these species against plant pathogenic *Fusarium* spp. [13,63,64]. The test that was performed in partitioned plates showed that VOCs are also involved in the antagonism of *T. afroharzianum* T138 against *Fg*. The antifungal activity of VOCs from endophytic *T. afroharzianum* strains against *Fusarium* spp. has recently been reported [65]. The production of fungal CW degrading enzymes by *Trichoderma* spp. is fully recognized [19,66] and their effect is synergistic with the antibiotic activity that is displayed by secondary metabolites against different soilborne phytopathogenic fungi [67]. In the present study, protease, cellulase, and chitinase activities were determined in extracellular protein extracts for the four *Trichoderma* strains after growing them in a synthetic medium containing glucose or *Fg* CWs. A two-way ANOVA showed that the values for the three enzymatic activities were affected by the variables strain and culture medium, and by the combination of both factors. In the presence of CWs of the pathogen, SM+0.5% *Fg*-CWs, strain T136 gave the highest cellulase and chitinase activity values. This result is consistent with its ability to overgrow *Fg* colonies, in comparison with the other three *Trichoderma* strains that were observed in the dual culture assay (Figure 2), and with the high inhibition percentages of the *Fg* growth obtained in the cellophane membrane assay (Table 2). The lack of cellulase activity that was observed in the supernatants from SM + 2% Glu cultures for the four strains shows a catabolite repression of cellulase genes. Although the production of cellulases depends on the microorganism, the carbon and nitrogen sources, and the cultivation conditions, it is well established that the use of glucose as a carbon source leads to a strong repression of cellulase gene transcription by this catabolite [68]. The chitinase activity that was detected in SM + 0.5% *Fg*-CWs for *T. harzianum* T136 could be a consequence of the presence of CWs of the pathogen in the culture medium since chitin is a major component of the CW of *Fg*. However, this was not the case for the other three *Trichoderma* strains.

The beneficial effects of *Trichoderma* on plants have been reported in terms of growth promotion and defense induction against biotic and abiotic stresses [23,25,69]. *Trichoderma* can modify the phytohormone signaling networks in plants [70]. Many *Trichoderma* strains produce IAA, a phytohormone that is crucial in controlling lateral root development and growth [24], as well as the enzyme ACCD, which is able to reduce the ET levels of the plant and as a result promotes growth [31]. *Trichoderma* spp. can also produce, to a greater or lesser extent, other phytohormones such as ABA, SA, GAs, and CKs [21,29,30,71]. We have previously found that SA production by *T. parareesei* positively affects its antifungal capacity against *Fusarium oxysporum* [21]. It may, therefore, be that the greater inhibition of *Fg* growth that was observed with *T. harzianum* T136 and T139 could be because they are able to produce more SA. It has been suggested that the production of the CK cZ by *Trichoderma* may be related to the ability to inhibit the growth of *Fg* [71]. We have also seen that the two strains that produced the CK tZ were also the ones that significantly inhibited the growth of *Fg*. Greenhouse assays that were performed on Basilio wheat with the four endophytic *Trichoderma* strains showed that all of them positively affected the plant emergence rate, tillering, and growth under optimal irrigation conditions. The significant differences in plant emergence rates that were observed at six days but not at nine days for the *Trichoderma* treatments compared to the control may indicate that these endophytic strains positively affect seed germination and plant earliness. Since these four endophytic *Trichoderma* strains presented ACCD activity and IAA production, their positive effect on wheat growth could be a consequence, at least in part, of these latter traits. In any case, plant growth promotion is not an inherent quality to the genus *Trichoderma*, as some strains have even been found to cause negative effects [55,72]. We have observed in another study that *Trichoderma* strains did not promote growth in wheat plants under optimal irrigation conditions, although some of them increased tolerance to drought [30].

In the present work, we also evaluated the effect of the four endophytic *Trichoderma* strains on wheat plants that were subjected to drought stress through stopping irrigation for 2.5 weeks and maintaining the plants until the crop was harvested. Except for T136 (Table 5), all plants from the other *Trichoderma* treatments showed, at 40 days, better fresh weight performance compared to the control when they were subjected to water stress. This result is consistent with observations over the long term since plants from the T136 treatment did not differ from the control plants in size and spike weight at 21 weeks (Figure 6). Interestingly, *T. harzianum* T136 showed the best biocontrol potential against *Fg* and performed worst in supporting plant growth traits under water stress conditions. Our greenhouse trial also showed that the combined application of the four endophytic *Trichoderma* strains did not improve the results that were achieved with some of the strains applied separately to optimally watered and water-stressed plants.

Plants overproduce ROS when exposed to drought or other environmental stresses. It is well established that plants possess an antioxidant system which involves the action of enzymes such as POD, SOD, and CAT, to reduce the damage that is associated with the overproduction of ROS [40]. It has been also reported that *Trichoderma* strains protect wheat plants against abiotic stresses by activating the ROS scavenging system [30,43,44]. In our study, we observed a higher H_2_O_2_ accumulation in drought-stressed plants, and that accumulation was higher in control plants than in those that were treated with *Trichoderma,* as previously described [30,73]. The decline in H_2_O_2_ accumulation is not exclusive to environmental stresses, as a decrease in H_2_O_2_ concentration has also been seen in *Trichoderma*-treated plants compared to those that were infected with fungal pathogens [10]. The levels of H_2_O_2_ were particularly low in water-stressed plants from the *T. simmonsii* T137 treatment (Figure 5), which also exhibited the highest SOD and POD activity values. Such increased levels of antioxidant activities have also been related to the behavior of tomato plants that were treated with *Trichoderma* after infection with *Fusarium oxysporum* f. sp. *lycopersici*, which is evidence that the maintenance of ROS homeostasis is a general plant defense mechanism that can be induced by *Trichoderma* [10]. In our study, the highest SOD and POD activity values that were detected in plants treated with T137 were accompanied by the best phenotypic performance of the water-stressed plants in the long term, suggesting that the T137 strain has the ability to help wheat plants overcome drought situations. In this sense, another *T. simmonsii* strain has been demonstrated to improve soybean seedling growth under conditions of optimal irrigation and drought [74]. The *T. simmonsii* T137 strain produced the highest amount of ABA and ACCD activity, and, together with *T. afroharzianum* T138, featured in the maximum IAA output. Previous works that were carried out in wheat have linked *Trichoderma* ACCD activity to increased plant tolerance to abiotic stresses, such as waterlogging [44] and drought [30]. In plants, ABA is a key phytohormone to regulate adaptation to unfavorable environmental conditions [75]. The ABA that is released by *Trichoderma* has been related to plant alleviation of salinity [76] and water stress [30]. IAA, in addition to controlling plant growth and root development, can increase plant tolerance to abiotic stresses, acting as a free radical scavenger and thereby overcoming oxidative stress [77]. A previous study has shown that the IAA that is produced by *T. longibrachiatum* contributed to the IAA concentration in wheat seedling roots under stress conditions [43]. Considering phytohormone production and results that were recorded in plants, *T. simmonsii* T137 has shown sufficient potential to make it worth assessing its practical applications in real field situations.

## 5. Conclusions

Despite the potential of beneficial fungi in agriculture, their role in the context of water deficits is poorly understood compared to their biocontrol abilities. Overall, this study demonstrated the usefulness of endophytic *Trichoderma* strains and their positive effects for use in agriculture. However, strain characterization and selection must be undertaken very carefully. The two strains of *T. harzianum* performed best as BCAs against *Fg*, but one of them, *T. harzianum* T136, showed the best fungal CW degrading enzyme activity, yet did not show any ability to protect the wheat plants from drought stress. *T. simmonsii* T137 was noted to be the best producer of the phytohormones ABA and IAA, and it had the highest ACCD activity. Under water deficit conditions, it was also the best strain to reduce plant-generated oxidative stress and to help wheat plants overcome drought stress.

## Figures and Tables

**Figure 1 jof-07-01087-f001:**
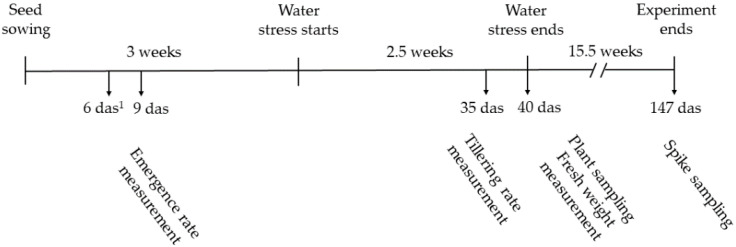
Representation of the running and sampling time points for the wheat-*Trichoderma* assay performed under greenhouse conditions. ^1^ das: days after sowing.

**Figure 2 jof-07-01087-f002:**
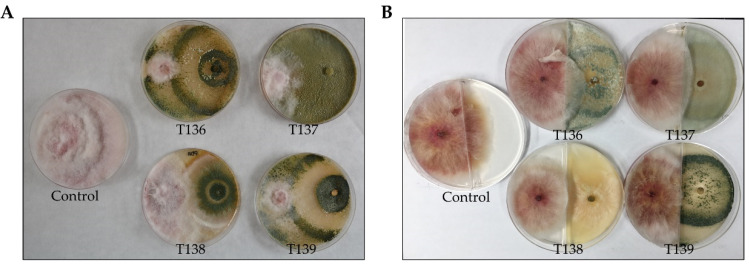
Dual cultures of four *Trichoderma* strains (*T. harzianum* T136, *T. simmonsii* T137, *T. afroharzianum* T138, and *T. harzianum* T139) (on the right of each plate) and the pathogen *Fusarium graminearum* (on the left). (**A**) Using 8.5 cm Petri plates, and (**B**) using centrally partitioned 8.5 cm Petri plates. Control corresponds to the pathogen growing alone. The plates were incubated at 28 °C for four days.

**Figure 3 jof-07-01087-f003:**
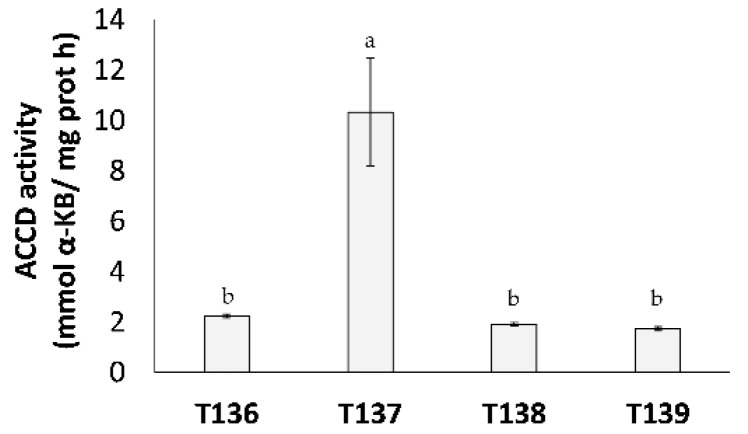
The 1-aminocyclopropane-1-carboxylic acid deaminase (ACCD) activity of four *Trichoderma* strains (*T. harzianum* T136, *T. simmonsii* T137, *T. afroharzianum* T138, and *T. harzianum* T139), expressed as millimoles of α-ketobutyrate in 1 h per milligram of protein, in a four day synthetic medium culture. The data were calculated from three replicates for each strain. Different letters above the bars indicate significant differences according to one-way analysis of variance (ANOVA) followed by Tukey’s test (*p* < 0.05).

**Figure 4 jof-07-01087-f004:**
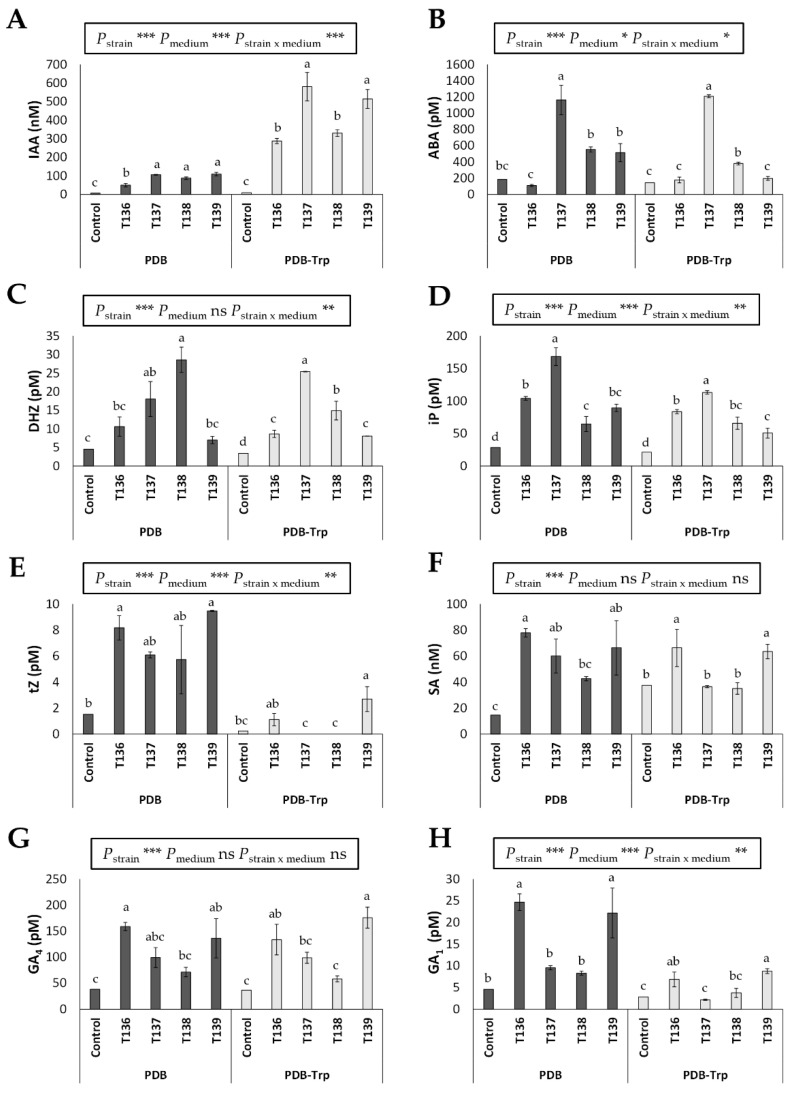
Phytohormone production in two day PDB and PDB-tryptophan (Trp) cultures by four *Trichoderma* strains (*T. harzianum* T136, *T. simmonsii* T137, *T. afroharzianum* T138, and *T. harzianum* T139) compared to their respective PDB and PDB-Trp media controls. (**A**) Indole-3-acetic acid (IAA), (**B**) abscisic acid (ABA), (**C**) cytokinin dihydrozeatin (DHZ), (**D**) cytokinin isopentenyladenine (iP), (**E**) cytokinin trans-zeatin (tZ), (**F**) salicylic acid (SA), (**G**) gibberellin 4 (GA_4_), and (**H**) gibberellin 1 GA1 (GA_1_). The data were calculated from three replicates per strain and culture medium. For each phytohormone and culture medium, different letters above the bars indicate significant differences according to one-way analysis of variance (ANOVA) followed by Tukey’s test (*p* < 0.05). For each phytohormone, significant effects were determined by a two-way ANOVA for *Trichoderma* strain, culture medium, and the combination strain per culture medium (***: *p* < 0.001; **: *p* < 0.001; *: *p* < 0.05; ns: no statistical differences).

**Figure 5 jof-07-01087-f005:**
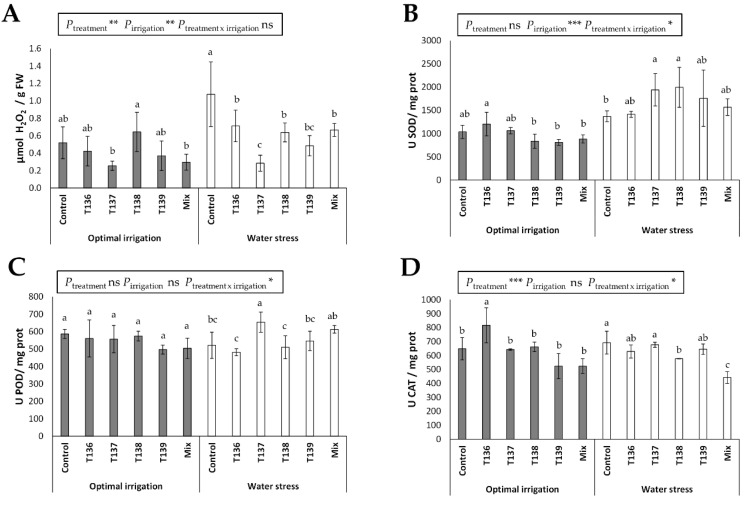
Effect of *Trichoderma* (*T. harzianum* T136, *T. simmonsii* T137, *T. afroharzianum* T138, *T. harzianum* T139, and a mixture with these four strains) treatments on (**A**) H_2_O_2_ content, and (**B**) SOD, (**C**) POD, and (**D**) CAT activities in 40-day-old wheat plants that were grown under optimal irrigation and water stress (irrigation was removed for 2.5 weeks when plants were 21 days old) conditions. The data were calculated from four replicates for each treatment and condition (*n* = 4), and different letters above the bars indicate significant the differences according to one-way analysis of variance (ANOVA) followed by Duncan’s test (*p* < 0.05). For each set of data, the effect of significance was determined by a two-way ANOVA for *Trichoderma* treatment, irrigation condition, and the combination treatment per irrigation condition (***: *p* < 0.001; **: *p* < 0.01; * *p* < 0.05; ns: no statistical differences). FW: leaf fresh weight.

**Figure 6 jof-07-01087-f006:**
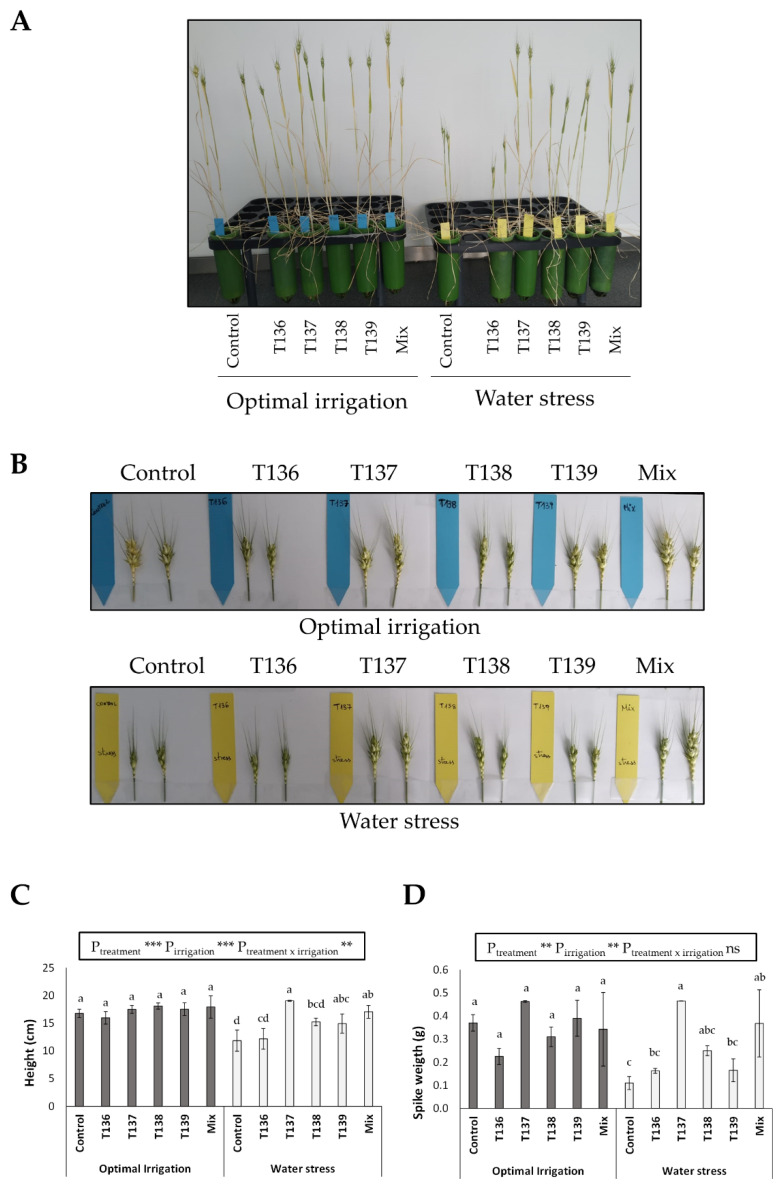
Long-term effect of *Trichoderma* (*T. harzianum* T136, *T. simmonsii* T137, *T. afroharzianum* T138, *T. harzianum* T139, and a mixture with these four strains) treatments on Basilio wheat under optimal irrigation and water stress (irrigation was removed for 2.5 weeks when the plants were 21 days old) conditions. Photographs of (**A**) plants and (**B**) spikes, taken at 5 months, and values of (**C**) plant height and (**D**) spike weight. Data in (**C**,**D**) were calculated from four replicates for each treatment and condition (*n* = 4), and the different letters above the bars indicate significant differences according to one-way analysis of variance (ANOVA) followed by Tukey’s test (*p* < 0.05). For each set of data, the effect of significance was determined by a two-way ANOVA for *Trichoderma* treatment, irrigation condition, and the combination treatment per irrigation condition (***: *p* < 0.001; **: *p* < 0.01; * *p* < 0.05; ns: no statistical differences).

**Table 1 jof-07-01087-t001:** Species assignment and accession numbers of *Trichoderma* strains isolated from the wheat endosphere in this study.

Strain	Identified as	ITS1-ITS4	*tef1α*	*acl1*
T136	*T. harzianum*	MT706636	MW803052	MW803056
T137	*T. simmonsii*	MT706637	MW803053	MW803057
T138	*T. afroharzianum*	MT706638	MW803054	MW803058
T139	*T. harzianum*	MW794309	MW803055	no data

**Table 2 jof-07-01087-t002:** Inhibition of *Fusarium graminearum* (*Fg*) colony growth by four *Trichoderma* strains (*T. harzianum* T136, *T. simmonsii* T137, *T. afroharzianum* T138, and *T. harzianum* T139) ^1^.

Strain	Dual Culture (%) ^2^	Cellophane Membrane (%) ^3^	Cellulose Membrane (%) ^3^
T136	90.3 ± 4.2 a	76.2 ± 0.4 a	53.8 ± 2.4 a
T137	58.1 ± 7.2 b	34.8 ± 0.1 b	22.6 ± 0.6 b
T138	40.5 ± 4.3 b	22.5 ± 8.5 b	16.4 ± 6.4 b
T139	85.2 ± 5.6 a	69.2 ± 1.4 a	49.3 ± 0.6 a

^1^ Results are expressed as the inhibition percentage with respect to the mean colony diameter of *Fg* grown alone. Values are the means of four replicates with the corresponding standard deviations. Different letters indicate differences within each column according to one-way analysis of variance (ANOVA) followed by Tukey’s test (*p* < 0.05). ^2^ Dual culture data were obtained after four-day growth on PDA. ^3^
*Fg* diameters were measured after three days of growth on the same PDA on which *Trichoderma* grew on cellophane or 12.5-kDa cut-off cellulose membrane for 36 h.

**Table 3 jof-07-01087-t003:** Protease, cellulase, and chitinase activities that were measured for four *Trichoderma* strains (*T. harzianum* T136, *T. simmonsii* T137, *T. afroharzianum* T138, and *T. harzianum* T139) in extracellular protein extracts from two culture media ^1^.

Strain	Synthetic Medium + 2% Glucose			Synthetic Medium + 0.5% *Fg*-Cell Walls		
	Protease ^2^	Cellulase ^2^	Chitinase ^2^	Protease ^2^	Cellulase ^2^	Chitinase ^2^
T136	175.3 ± 43.5 a	0.0	1.4 ± 0.2 b	93.8 ± 5.7 a	7.8 ± 0.6 a	24.0 ± 5.6 a
T137	136.4 ± 7.9 ab	0.0	5.5 ± 0.9 ab	39.1 ± 8.6 b	2.6 ± 0.2 c	7.0 ± 0.4 bc
T138	0.0 c	0.0	13.1 ± 5.1 a	86.6 ± 5.5 a	3.3 ± 0.3 c	14.1 ± 0.8 b
T139	78.6 ± 18.4 b	0.0	8.2 ± 3.7 ab	91.0 ± 9.3 a	5.5 ± 1.1 b	4.1 ± 0.4 c

^1^ Extracellular protein was obtained from five day liquid cultures in a synthetic medium supplemented with 2% glucose or 0.5% *Fusarium graminearum* (*Fg*) cell walls. ^2^ The data are expressed in micromoles of azocasein (protease activity), carboxymethylcellulose (cellulase activity), and N-acetylglucosamine (chitinase activity) in 1 min per milligram of protein. The values are the means of three replicates with the corresponding standard deviations. Different letters indicate the differences within each column according to one-way analysis of variance (ANOVA) followed by Tukey’s test (*p* < 0.05).

**Table 4 jof-07-01087-t004:** Root colonization of wheat plants of the variety Basilio by four *Trichoderma* strains (*T. harzianum* T136, *T. simmonsii* T137, *T. afroharzianum* T138, and *T. harzianum* T139) ^1^.

Strain	*Trichoderma* Actin		Wheat Actin		Ratio ^5^
	Ct ^2^	Qyt (ng) ^3^	Ct ^2^	Qty (ng) ^4^	
Uninoculated	29.45 ± 0.14	-	22.11 ± 0.29	4.02 ± 0.54	0.0 b
T136	14.62 ± 0.61	9.36 ± 1.08	22.48 ± 0.28	3.32 ± 0.52	2.89 ± 0.78 a
T137	14.53 ± 0.22	9.51 ± 0.40	22.70 ± 0.42	2.92 ± 0.77	3.41 ± 0.92 a
T138	14.74 ± 0.89	9.14 ± 1.58	22.72 ± 0.25	2.89 ± 0.45	3.24 ± 0.93 a
T139	15.55 ± 0.41	7.71 ± 0.73	22.96 ± 0.55	2.45 ± 1.01	3.37 ± 1.09 a
Mix	16.42 ± 0.48	6.16 ± 0.84	23.41 ± 0.25	1.63 ± 0.46	3.86 ± 0.57 a

^1^ Fungal DNA was quantified by qPCR. The data were calculated from four replicates per strain, and the values presented as means with the corresponding standard deviations. ^2^ Ct, threshold cycle. ^3^ Quantity of *Trichoderma* DNA (ng), referred to as *Trichoderma* actin gene, -: negative values. ^4^ Quantity of wheat DNA (ng), referred to as wheat actin gene. ^5^ Proportion of fungal DNA vs. plant DNA; values with different letters are significantly different according to one-way analysis of variance (ANOVA) followed by Tukey’s test (*p* < 0.05).

**Table 5 jof-07-01087-t005:** Effect of five *Trichoderma* treatments (*T. harzianum* T136, *T. simmonsii* T137, *T. afroharzianum* T138, *T. harzianum* T139, and a mixture with these four strains) on emergence, tillering, and fresh weight of Basilio wheat grown in a greenhouse under optimal irrigation and water stress (irrigation was removed for 2.5 weeks when plants were 21 days old) conditions.

Treatment	Emergence Rate (%) ^1^		Tillering Rate (%) ^2^		Fresh Weight (g) ^3^	
	6 Das	9 Das	Optimal Irrigation	Water Stress	Optimal Irrigation	Water Stress
Control	46.7 ± 13.9 c	76.7 ± 19 a	16.7 c	25.0 c	0.64 ± 0.08 c	0.60 ± 0.08 c
T136	80.0 ± 7.5 a	93.3 ± 9.1 a	66.7 b	0.0 d	0.94 ± 0.15 ab	0.61 ± 0.10 bc
T137	83.3 ± 11.8 a	96.7 ± 7.5 a	66.7 b	50.0 c	0.99 ± 0.06 a	0.80 ± 0.09 ab
T138	76.7 ± 14.9 ab	93.3 ± 9.1 a	83.3 ab	100.0 a	1.15 ± 0.16 a	0.91 ± 0.14 a
T139	63.3 ± 7.45 bc	83.3 ± 16.7 a	66.7 b	40.0 c	0.95 ± 0.09 ab	0.81 ± 0.12 ab
Mix	70.0 ± 13.9 b	90 ± 14.9 a	90.0 a	70.0 b	1.20 ± 0.16 a	0.82 ± 0.05 a

^1^ Data taken at six and nine days after sowing are expressed in percentages and were calculated in 16 replicates per treatment (*n* = 16). ^2^ Data that were recorded in 35-day-old plants are expressed in percentages and were calculated in eight replicates per treatment and condition (*n* = 8). ^3^ Data that were recorded in 40-day-old plants are expressed in g and were calculated in eight replicates per treatment and condition (*n* = 8). All the values are means with the corresponding standard deviations. Different letters indicate differences within each column according to one-way analysis of variance (ANOVA) followed by Tukey’s test (*p* < 0.05).

## Data Availability

Not applicable.

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
