# Peer review of "Why Is the Correct Selection of Trichoderma Strains Important? The Case of Wheat Endophytic Strains of T. harzianum and T. simmonsii"

_jof, 2021, doi:10.3390/jof7121087_

Round 1

Reviewer 1 Report

The manuscript presents the results of the selection of endophytic fungi from wheat plants for their use in crops to improve plant growth  under stress conditions. However, my main comment relates to the final conclusions regarding the performance of the tested Trichoderma strains under stress conditions. In this paper, the study of the abilities of isolated Trichoderma is divided into two stages: (i) in vitro assays against phytopathogenic fungus Fusarium graminearum (Fg); (ii) in vivo trials with wheat plants to check the performance of Trichoderma under water stress. According to the results T136 exhibited the best properties to control Fg, and T137 to alleviate drought stress. However, the conclusions suggest, to my thinking, that strain T136, despite its activity against the pathogen, is 'inferior' to T137, because it did not protect the wheat from drought stress. Is this really the case? The in vivo studies are somewhat lacking in variants, where the pathogen should be present, to check the actual effectiveness of the tested strains in both cases: pathogen, drought. I miss the relationship between in vitro studies relating to activity against the pathogen and in vivo with drought stress. 

The second question refers to Trichoderma mixture. How the Authors know that all Trichoderma applied in the mixture colonised wheat, and to what extent? In the table 4 there is only information about the quantity of Trichoderma DNA in plant material in general, wihout subdivision into strains. Therefore, it is not known whether the strains colonised evenly or whether some were predominant.

In conclusion, the manuscript is very well prepared and is very informative. The scientific merit is average, due to the numerous related works, but this does not reduce the value of the work in general.

Author Response

Many thanks for your comments which have helped to improve the quality and presentation of the article. 

The manuscript presents the results of the selection of endophytic fungi from wheat plants for their use in crops to improve plant growth under stress conditions. However, my main comment relates to the final conclusions regarding the performance of the tested Trichoderma strains under stress conditions. In this paper, the study of the abilities of isolated Trichoderma is divided into two stages: (i) in vitro assays against phytopathogenic fungus Fusarium graminearum (Fg); (ii) in vivo trials with wheat plants to check the performance of Trichoderma under water stress. According to the results T136 exhibited the best properties to control Fg, and T137 to alleviate drought stress. However, the conclusions suggest, to my thinking, that strain T136, despite its activity against the pathogen, is 'inferior' to T137, because it did not protect the wheat from drought stress. Is this really the case?

--- Answer:  Sure, as it can be seen in Tables 2 and 3, strain T136 was the best strain when the criterion was biocontrol ability. However, if we look at the results in Figure 6, plants challenged with this strain did not differ from control plants in tests under drought stress conditions, as did plants treated with strain T137. This is why we want to emphasize in this paper that it is so important to make a correct selection of Trichoderma strains.

The in vivo studies are somewhat lacking in variants, where the pathogen should be present, to check the actual effectiveness of the tested strains in both cases: pathogen, drought. I miss the relationship between in vitro studies relating to activity against the pathogen and in vivo with drought stress. 

--- Answer: Surely this approach would be ideal, but we also think that it would be very difficult to draw conclusions given the complexity of the system. We know that plant responses to Trichoderma vary over time and that the expression of genes governing the defense pathways is undulating. The introduction of a pathogen into the system depends on its concentration, and its effects can be more or less rapid. If we make disease symptoms coincide with drought symptoms, water stress may also affect pathogen growth and virulence. In the present work we wanted to select strains for their potential antagonistic mechanisms and for their action on plants under drought conditions and therefore we have tried to simplify the system. In any case, work with the individual strains is not finished and the suggestion of Reviewer 2 will be taken into account.

The second question refers to Trichoderma mixture. How the Authors know that all Trichoderma applied in the mixture colonised wheat, and to what extent? In the table 4 there is only information about the quantity of Trichoderma DNA in plant material in general, wihout subdivision into strains. Therefore, it is not known whether the strains colonised evenly or whether some were predominant.

--- Answer: We performed this experiment but did not include it in the previous version of the manuscript. We have seen that the four Trichoderma strains colonized evenly. This new information and a new reference (Elad et al., 1981) have been included in the current version:

--- M&M (new lines 228-237): “Quantification of Trichoderma strains was also carried out by counting the number of colony-forming units (cfu) from one root (ca. 0.25 g) of each wheat plant. For this, roots were sampled on wheat plants of the greenhouse assay described below at 21 days after sowing (das). Each sampled root was washed twice in 20 mL PBS-S buffer by shaking at 180 rpm in a 50 mL tube for 20 min. Serial dilutions of the rhizosphere washing liquid were plated on Trichoderma selective medium (TSM) [57]. The plates were incubated at 28 ºC and the cfu were counted after six days. In addition, the colonies of the Mix treatment that grew on TSM were then separately plated on PDA for subsequent morphological distinction and Trichoderma species assignment. One root of four wheat plants per treatment was tested”.

--- Results (new lines 445-451): “The abundance of Trichoderma strains in 21 das samples was 2.4 x 105, 0.7 x 105, 4.3 x 105 and 4.9 x 105 cfu per wheat plant root for strains T136, T137, T138, and T139, respectively. No Trichoderma colonies were observed in the roots of the control treatment. Regarding the Mix treatment, Trichoderma abundance was found to be in a similar range, 1 x 105, to individual inoculations. An analysis of the count data showed no differences among Trichoderma treatments. In addition, considering the colony phenotypes on PDA, the four strains had similar percentage counts in the Mix treatment”.

--- Discussion (new lines 569-574): “qPCR colonization data obtained 48 h after 10-day-old seedling inoculation with Trichoderma in liquid medium were in line with those obtained by assessing Trichoderma survival in 3-week-old plants grown under greenhouse conditions. Data obtained in both experiments for individual inoculations and the Mix treatment showed an absence of negative interactions among the four Trichoderma strains, indicating compatibility when mixed”.

Reviewer 2 Report

It is an interesting work. I have some suggestions for the authors for further improvement of current version:

Introduction: hypothesis part should be improved

What was the rationale behind selection of PEG concentrations?

Method for 2.2. Fungal Molecular Identification can be elaborated more for better readership.

Provide primer information for qRT-PCR .

Table 3, provide units of enzyme activity, see for other tables as well.

Please increase font size in fig 4

Discussion is fine, but I suggest improving discussion related to antioxidant system

Author Response

Many thanks for your omments which have helped to improve the quality and presentation of the article. 

---- As you indicated, the current version of the manuscript has been English checked.

Introduction: hypothesis part should be improved

--- Done. This is the new text (new lines 90-93, and 97-98): “… Our aim was to determine whether phylogenetically very close Trichoderma endophytic isolates show similar biocontrol potential and/or beneficial effects on wheat plants, in order to select them for the most efficient application. To achieve this goal, the four Trichoderma strains were …” and “… vi) ability to alleviate drought stress in wheat plants”.

What was the rationale behind selection of PEG concentrations?

--- Polyethylene glycol (PEG) simulates drought stress. As it can be seen in Table S1, we carried out a sliding scale of PEG values in ten-by-ten percentage steps. However, 40% PEG is equivalent to a water potential of -1.76 MPa.

---  We have written this new text in Result section (new lines 304-309): “This percentage of PEG is equivalent to a water potential of -1.76 MPa, and although some isolates were affected by this simulated drought stress, it was not a constant for all fungi tested. Moreover, no correlation was found between belonging to a fungal genus and growing at a given PEG concentration, suggesting an isolate-dependent property. The four isolates of the genus Trichoderma that grew to different extents in PDB plus 40% PEG were chosen for further testing”.

Method for 2.2. Fungal Molecular Identification can be elaborated more for better readership.

--- Done. This is the new text and a new reference (Jaklitsch and Voglmayr, 2015) have been included in M&M section (new lines 127-140): “Total fungal DNA was extracted following the method of Raeder and Broda [46], using mycelium collected from a cellophane sheet deposited on the surface of a PDA plate, where the fungus was grown at 28 ºC for 48 h. The ITS1-ITS4 region of the nuclear rDNA gene cluster, including ITS1 and ITS2 and the 5.8S rDNA gene, was amplified with the primer pair ITS1//ITS4 for 54 fungal isolates, as previously described [17]. In addition, a fragment of the tef1a gene and one fragment of the acl1 gene were respectively amplified with the primer pairs EF1-728F (5’-CATCGAGAAGTTCGAGAAGG-3’)//tef1rev (5’-GCCATCCTTGGAGACCAGC-3’) and acl1-230up (5’-AGCCCGATCAGCTCATCAAG-3’)//acl1-1220low (5’-CCTGGCAGCAAGATCVAGGAAGT-3’) for the four Trichoderma isolates, as previously described [47,48]. PCR products were electrophoresed on 1% agarose gels. Amplicons were excised from the gels, purified, and sequenced as previously described [17,30]. The sequences obtained were analyzed considering homology in the NCBI database, with ex-type strains and taxonomically established isolates of Trichoderma as references”.

Provide primer information for qRT-PCR .

--- Done. This text has been added (new lines 220-223): “… Act-F (5’-ATCGGTATGGGTCAGAAGGA-3’)//Act-R (5’-ATGTCAACACGAGCAATGG-3’) [56] and Act-Fw (5’-TGACCGTATGAGCAAGGAG-3’)//Act-Rv (5’-CCAGACACTGTACTTCCTC-3’) …”.

Table 3, provide units of enzyme activity, see for other tables as well.

--- Done. This sentence has been included in Table 3 footer text we have written this (new lines 379-381): “Data are expressed in micromoles of azocasein (protease activity), carboxymethylcellulose (cellulase activity), and N-acetylglucosamine (chitinase activity) in 1 minute per milligram of protein”.

--- Inside Tables 2, 4 and 5, units (g or ng) or percentages (%) are now specified as appropriate.

Please increase font size in fig 4

--- Done. A new Figure 4 has been added.

Discussion is fine, but I suggest improving discussion related to antioxidant system

--- Done. These two paragraphs have been included in the Discussion section (new lines 652-656 and 658-663): “In our study, we observed a higher H2O2 accumulation in drought-stressed plants, and that accumulation was higher in control plants than in those treated with Trichoderma, as previously described [30,73]. The decline in H2O2 accumulation is not exclusive to environmental stresses, as a decrease in H2O2 concentration has also been seen in Trichoderma-treated plants compared to those infected with fungal pathogens [10].”, and “Such increased levels of antioxidant activities have also been related to the behavior of tomato plants treated with Trichoderma after infection with Fusarium oxysporum f. sp. lycopersici, which is evidence that the maintenance of ROS homeostasis is a general plant defense mechanism that can be induced by Trichoderma [10]. In our study, the highest SOD and POD activity values detected in plants treated with T137 …”.

Reviewer 3 Report

The present manuscript submitted for review includes information about the wheat endophytic strains of Trichoderma harzianum and Trichoderma simmonsii.

Authors this manuscript confirmed that the two strains of Trichoderma harzianum performed best as biological control agent against the phytopathogenic fungus Fusarium graminearum.

In general terms the topic of the reviewed article is interesting.

The  manuscript was prepared with care and its content contains a lot of valuable information. The work does not raise any scientific or substantive reservations.

The manuscript is well structured, the methodology is explicitly presented and the results reported are interesting.

The structure of the paper is correct. The tables and figures clearly presenting the obtained results with their appropriate interpretation. The statistical calculation methods used in the research make the obtained results reliable and provide a basis for drawing correct conclusions.

The paper needs some editorial corrections, after which it should be accepted in print.

I think that the paper can be published in the Journal of Fungi.

Author Response

The present manuscript submitted for review includes information about the wheat endophytic strains of Trichoderma harzianum and Trichoderma simmonsii. Authors this manuscript confirmed that the two strains of Trichoderma harzianum performed best as biological control agent against the phytopathogenic fungus Fusarium graminearum.

In general terms the topic of the reviewed article is interesting. The manuscript was prepared with care and its content contains a lot of valuable information. The work does not raise any scientific or substantive reservations. The manuscript is well structured, the methodology is explicitly presented and the results reported are interesting.

The structure of the paper is correct. The tables and figures clearly presenting the obtained results with their appropriate interpretation. The statistical calculation methods used in the research make the obtained results reliable and provide a basis for drawing correct conclusions. The paper needs some editorial corrections, after which it should be accepted in print. I think that the paper can be published in the Journal of Fungi.

--- Many thanks for the comments and the positive consideration of our work.

Round 2

Reviewer 2 Report

Authors have addressed all of my comments